# A Novel Object-Level Building-Matching Method across 2D Images and 3D Point Clouds Based on the Signed Distance Descriptor (SDD)

Chunhui Zhao [1,2], Wenxuan Wang [1,2], Yiming Yan [1,2,*], Nan Su [1,2], Shou Feng [1,2], Wei Hou [3] and Qingyu Xia [1,2]

[1] Key Laboratory of Advanced Marine Communication and Information Technology, Ministry of Industry and Information, Harbin 150009, China; zhaochunhui@hrbeu.edu.cn (C.Z.); wangwenxuan@hrbeu.edu.cn (W.W.); sunan08@hrbeu.edu.cn (N.S.); fengshou@hrbeu.edu.cn (S.F.); xiaqingyu@hrbeu.edu.cn (Q.X.)
[2] College of Information and Communication Engineering, Harbin Engineering University, Harbin 150009, China
[3] Harbin Aerospace Star Data System Science and Technology Co., Ltd., Harbin 150028, China; houwei@spacestar.com.cn
* Correspondence: yanyiming@hrbeu.edu.cn

**Abstract:** In this work, a novel object-level building-matching method using cross-dimensional data, including 2D images and 3D point clouds, is proposed. The core of this method is a newly proposed plug-and-play Joint Descriptor Extraction Module (JDEM) that is used to extract descriptors containing buildings' three-dimensional shape information from object-level remote sensing data of different dimensions for matching. The descriptor is named Signed Distance Descriptor (SDD). Due to differences in the inherent properties of different dimensional data, it is challenging to match buildings' 2D images and 3D point clouds on the object level. In addition, features extracted from the same building in images taken at different angles are usually not exactly identical, which will also affect the accuracy of cross-dimensional matching. Therefore, the question of how to extract accurate, effective, and robust joint descriptors is key to cross-dimensional matching. Our JDEM maps different dimensions of data to the same 3D descriptor SDD space through the 3D geometric invariance of buildings. In addition, Multi-View Adaptive Loss (MAL), proposed in this paper, aims to improve the adaptability of the image encoder module to images with different angles and enhance the robustness of the joint descriptors. Moreover, a cross-dimensional object-level data set was created to verify the effectiveness of our method. The data set contains multi-angle optical images, point clouds, and the corresponding 3D models of more than 400 buildings. A large number of experimental results show that our object-level cross-dimensional matching method achieves state-of-the-art outcomes.

**Keywords:** cross-dimensional remote sensing data; object-level matching; 2D optical image; 3D point cloud

## 1. Introduction

Building matching (BM) is the task of determining the corresponding data of buildings in query data from a database with given geographic labels and other information, which can be applied to the real-time positioning of unmanned aerial vehicles (UAV) [1–3], visual pose estimation [4–6], and 3D reconstruction [7] in remote sensing. The most common data type used for BM is images. Image-based retrieval methods for BM (e.g., [8–11]) attempt to identify similar database images that depict the same landmarks as the query image. Typically, the retrieved images are ranked according to a given similarity metric (e.g., the L1 norm distance between Bag-of-Words vectors [12], L2 norm distance between the compact representation and vector [13,14]) to obtain the BM result. In localization tasks, the position of the best-matching database image is usually treated as the position of the query image, or

the positions of the top N images are fused to obtain the position of the query image [15–17]. With the rapid development of deep learning, various end-to-end deep methods have been studied, significantly improving the accuracy of BM.

Although the above methods have led to achievements, it is well-understood that their BM accuracy is largely dependent on the quality and quantity of images in the database. Specifically, if the images of the corresponding area in the database have significant differences from the matching image in terms of shooting angles and lighting conditions, accurate BM cannot be achieved in the given area. The images in the database are downloaded in bulk from the network. On the one hand, the number of images of tourist attractions or landmarks in the network is much higher than that of ordinary buildings, which leads to a highly uneven distribution of BM accuracy. When the building image to be matched appears infrequently in the network, it is not easy to obtain accurate matching results. On the other hand, for large-scale maps [18], this method requires a large number of images in the database as a support, which requires high computing power and storage resources. When the target carrier of the localization task is an unmanned aerial vehicle that has lost network control signals, it will be difficult to complete BM without the support of ground data resources and computing resources. Compared with two-dimensional image data, three-dimensional point cloud data composed of coordinates do not have the problem of the shooting angle and light conditions and do not require the storage a large amount of image data in the database. Therefore, compared with the image-matching positioning method, the image-point-cloud-matching method requires less computing power and storage resources at the terminal. Secondly, the image-point-cloud-matching method is not affected by the quality and quantity of images in the database, reducing the possibility of erroneous matching caused by the problem of data imbalance in the database. Therefore, in order to overcome the serious dependence of the localization task on the image database and reduce the demand for computing and storage resources for BM-based localization tasks, the pixel-level cross-dimensional matching method for 2D optical images and 3D point clouds provides a promising strategy, as shown in Figure 1a.

The current cross-dimensional data pixel-level matching methods can be divided into three categories. The typical process of the first category is to first recover the 3D structure of the scene [19,20], usually reconstructed from images taken at equal intervals using motion structure (SfM) [21,22] and multi-view systems (MVS) [23] as the input. Each 3D point is triangulated using multiple 2D local features (such as SIFT [24]) and associated with the corresponding image descriptor. Then, the pixel-level cross-dimensional correspondence between the local feature descriptor in the query image and the 3D point descriptor is found [15]. These pixel-level descriptors are homogeneous, because the points inherit the descriptors of the corresponding pixels in the reconstructed 3D scene [25]. The second category of methods identifies associations between different-dimensional data by mapping descriptors from different domains to a shared latent space. However, they only construct block-by-block descriptors, which typically lead to coarse-grained matching results. Instead, 3DTNet [26] takes 2D and 3D local patches as the input and extracts 2D features from 2D patches using unique 3D descriptors that help it to learn 3D patches. However, 3DTNet is only used for 3D matching. The network uses 2D features as auxiliary information to render 3D features more discriminative. Recently, learning descriptors that allow for direct matching and retrieval between 2D and 3D local patches have been proposed with 2D3DMatch-Net [27] and LCD [28]. Other research has established a connection between 2D and 3D features for specific applications, such as object pose estimation. Additionally, some methods achieve cross-dimensional matching through registration, such as that described in [29], by converting registration problems into classification and inverse camera projection optimization problems using relative rigid transformation for cross-dimensional matching. The authors of [30] proposed a method that uses a two-stage approach to align two inputs of data in a virtual reference coordinate system (virtual alignment) and then compare and align the data to complete the matching. However, even if accurate descriptors can be extracted from 2D images and 3D point clouds, the above two categories of methods

still cannot establish accurate pixel-level cross-dimensional matching relationships. There are two reasons for this phenomenon. Firstly, due to the sparsity of point clouds, local point descriptors can be mapped to many pixel descriptors in 2D images, increasing matching ambiguity. To address this problem, Liu et al. [30] proposed a large-scale camera positioning method based on 2D–3D data matching, adding a disambiguation module that uses global contextual information to solve the problem of matching ambiguity. Secondly, as 2D images are usually a 2D mapping of the appearance of the scene [31], while 3D point clouds encode the structure of the scene, there are significant differences between the attributes of 2D images and 3D point clouds, and the descriptor loss functions of existing 2D or 3D local feature descriptions [32–34] cannot achieve accurate convergence in cross-dimensional matching tasks. Therefore, it is important to perform object-level cross-dimensional data matching for more effective solutions to the above problems.

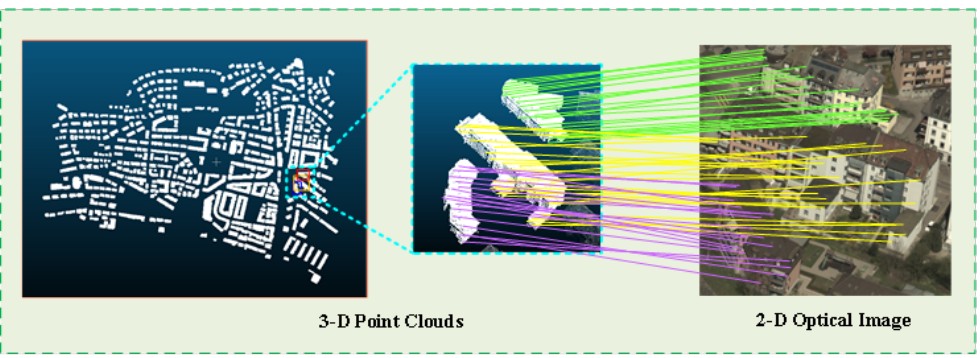

(a) Cross-dimensional pixel-level buildings matching

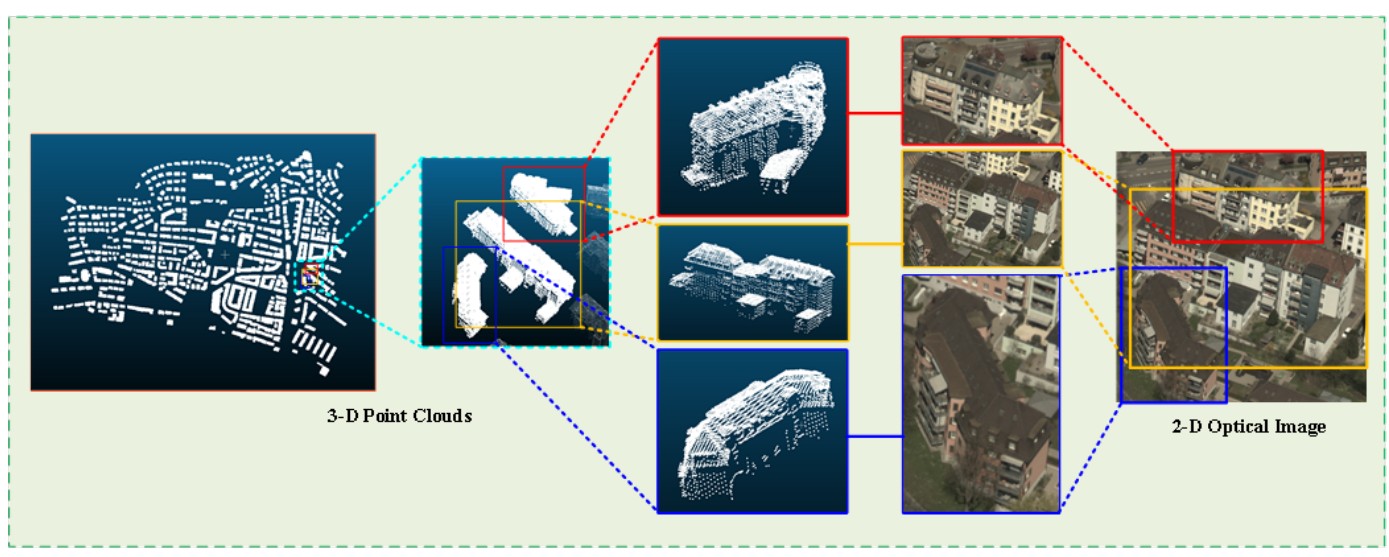

(b) Cross-dimensional object-level buildings matching

**Figure 1.** Comparative diagram of pixel-level cross-dimensional matching and object-level cross-dimensional matching. Pixel-level cross-dimensional matching takes pixels in the image and points in the point cloud as the core for matching, while object-level cross-dimensional matching takes the building objects in the image and the building objects in the point cloud as the core for matching.

In the task of matching building images and point clouds, object-level cross-dimensional data-matching methods match the building objects in the images and the point clouds as the core, as shown in Figure 1b. Compared with pixel-level matching methods, object-level matching methods extract joint descriptors from the data containing global information of the targets and map them to the descriptor space for matching. The global feature extraction method, which focuses on the overall target effectively, alleviates the problem of fuzzy

matching caused by the sparsity of point clouds, and the descriptor space solves the problem of attribute differences in cross-dimensional data, reducing the dependence of traditional image-based positioning methods on large amounts of image data. This concept provides new ideas for various fields, such as the precise self-positioning of drones in the state of control signal loss, urban management, smart city construction, high-quality building shape reconstruction, and so on. The core of this method lies in the choice of the descriptor space. By choosing a better descriptor space, one can obtain more accurate cross-dimensional matching results. The authors of [35] proposed a joint global embedding method for 3D shapes and images to solve retrieval tasks. By directly binding handcrafted 3D descriptors to learned image descriptors, cross-dimensional descriptors for object-level retrieval tasks were generated. The authors of [36] proposed a deep-learning-based cross-dimensional object-level descriptor space occupancy probability descriptor (SOPD) which uses the occupancy probability of each unit space of the object as the cross-dimensional descriptor.

In this study, we designed a plug-and-play Joint Descriptor Extraction Module (JDEM) to extract our proposed new joint descriptors, called Symbolic Distance Descriptors (SDD). The SDD utilize the 3D geometric invariance of building objects, using features that contain their 3D structural information as descriptors. This not only overcomes the inherent differences in attributes between 2D and 3D data but also reduces the fuzzy matching caused by the similarity between pixel-level descriptors. In addition, we propose Multi-View Adaptive Loss (MAL) to improve the adaptability of the image encoder module to images taken from different angles and to enhance the robustness of the joint descriptors. To achieve cross-dimensional BM, a corresponding database is required. Although Yan et al. [36] proposed a 2D–3D object-level building-matching data set, only its 3D point clouds were obtained from the real world. We constructed a fully real-world 2D–3D cross-dimensional object-level building-matching data set called 2D-3D-CDOBM. The data set contains multi-angle optical images, point clouds, and the corresponding 3D models for over 400 buildings. We conducted extensive experiments on our data set to verify that our proposed descriptors can accurately perform cross-dimensional object-level matching tasks.

The important contributions of this paper are as follows:

- A novel cross-dimensional object-level BM method based on the SDD is proposed. The method compensates for the modal differences between 2D optical images and 3D point clouds in object-level building data.
- A plug-and-play JDEM is proposed. JDEM utilizes the three-dimensional geometric invariance of objects in the real physical world to obtain the same domain features by mapping the different dimensional data of objects to a three-dimensional space.
- MAL is proposed to reduce the distance of descriptors extracted from images of the same object from different angles in the SDD space. The loss function improves the adaptability of the image encoder module to images from different angles.
- A cross-dimensional object-level building-matching data set composed of real data, named 2D-3D-CDOBM, is proposed to verify the effectiveness of our method. It includes optical images of buildings taken from different angles, LiDAR point clouds of buildings scanned in different years, and hundreds of corresponding 3D models of buildings. This data set is available to download at "https://github.com/HEU-super-generalized-remote-sensing/Cross-dimensional-Object-level-Matching-Data-Set.git (accessed on 4 June 2023)".

## 2. Materials and Methods

This section describes the details of the proposed descriptor, SDD (Section 2.1), the algorithm structure of the proposed cross-dimensional matching method (Section 2.2), the used loss function (Section 2.3), the proposed data used for the training, verification, and testing of the algorithm (Section 2.4), evaluation metrics (Section 2.5), and the training platform and parameter settings (Section 2.6).

## 2.1. SDD

Under ideal conditions, the imaging mechanism of 2D images can be simplified to a pinhole imaging model in which the captured object is projected onto the photosensitive element through the pinhole. Therefore, the images can be regarded as the projection of the 3D world scene to the 2D space. The image data format is $h \times v$, where $h$ is the number of horizontal pixels, and $v$ is the number of vertical pixels. However, in the projection process, the loss of information is likely to occur due to the positional relationship, such as the incomplete structure of the occluded object in the image caused by the occlusion relationship.

While 3D point clouds are usually obtained via LiDAR scanning, by emitting a laser beam towards an object, LiDAR receives the laser radiation reflected by the scene and produces a continuous analog signal. At last, LiDAR restores this to a point cloud of the object scene [37,38]. The data form of the point cloud is shown in (1):

$$p_i = (x, y, z)$$
$$P = \{p_1, p_2, p_3, p_4, \cdots\} \tag{1}$$

where $x$, $y$, and $z$ are the relative position coordinates of the points $p_i$ in the point cloud in the world's physical coordinate system. LiDAR is an active sensor, and the number of point clouds generated by it is positively correlated with the LiDAR scanning time. The point density at the top of the building is higher than that on the sides of the building. Therefore, the matching of the 2D images and the 3D point clouds requires effort.

However, neither 2D images nor 3D point clouds are naturally generated. They are both derived from the 3D physical world via mapping or projection through corresponding sensors. Although the data formats are different, both are embodiments of the 3D physical world object containing the corresponding 3D information, that is, the 3D geometric invariance. According to this characteristic, by extracting 3D geometric descriptors from 2D images and 3D point clouds, one can repair their missing information and map descriptors extracted from different dimensional data to the same descriptor space. Therefore, the key to the cross-dimensional matching method is descriptor extraction.

Our method uses the SDD as a cross-domain descriptor extracted from different dimensional data. The SDD is a descriptor obtained via sampling of the Symbol Distance Feature (SDF) [39] at the threshold $\tau$. SDF is a three-dimensional feature that describes the distance from any point in the normalized metric space to the model boundary, as shown in Figure 2b. If the sampling point is within the boundary, its SDF feature for the current sampling point is set to negative. Conversely, if the point is outside the boundary, its SDF feature is set to positive. The farther the sampling point is from the boundary, the greater the modulus of SDF will be. Since SDF is a description of distance, the threshold $\tau$ is usually set to zero.

Similar to the SDD, [36] the Spatial Occupancy Probability Feature (SOPF) is used to describe the three-dimensional structure of the building, that is, the probability of the current sampling point being inside the model. Therefore, the SOPF of the sampling points outside the model surface is set to 0, and that inside is set to 1 (100%), as shown in Figure 2c. Similarly, the SOPD also uses the threshold $\tau_1$ to sample the SOPF, and the threshold $\tau_1$ is usually set to 0.5 (50%). Compared with classification problems, deep learning networks have a better adaptability to regression problems. Therefore, compared with the SOPD, the network has a more significant learning ability and better learning effect for the SDD, which is also reflected in the experimental section (Section 3).

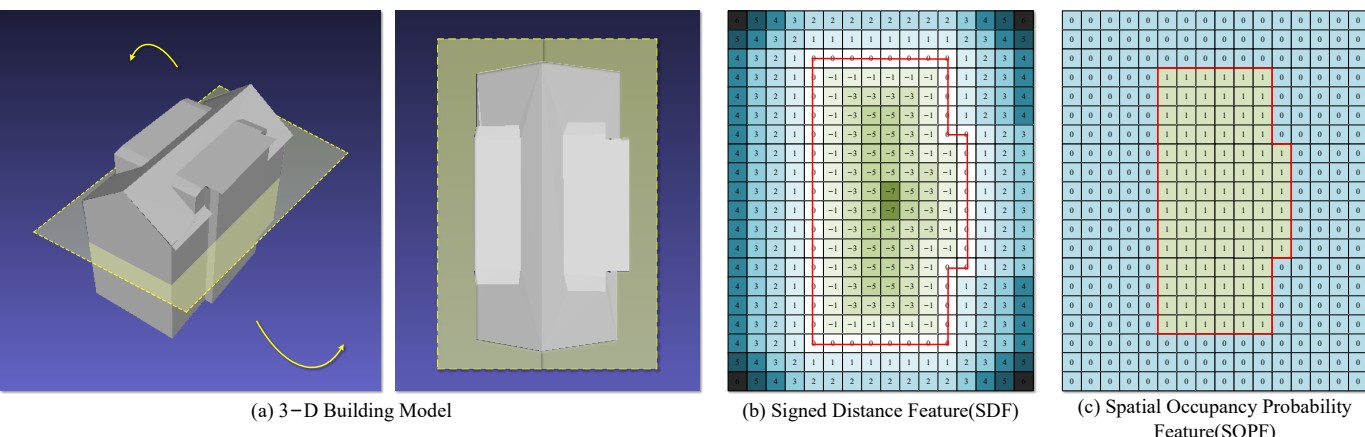

(a) 3−D Building Model
(b) Signed Distance Feature(SDF)
(c) Spatial Occupancy Probability Feature(SOPF)

**Figure 2.** The diagram of the SDF and SOPF. The red line in (**b**,**c**) represents the boundary of the building in (**a**), green represents the interior of the boundary, while blue represents the exterior of the boundary, and the darker the color in different positions, the larger the descriptor value in that position.

### 2.2. Cross-Dimensional Matching Method

There are many mature image feature extraction methods, such as VGG [40] and ResNet [41], and many mature point cloud feature extraction methods, such as PointNet [42] and PointNet++ [43]. In deep learning, the above encoder completes the front-end feature extraction function. The mapping relationship between the data and feature vectors of different modes is established. At this time, the formal unification task for different modal data features is completed. However, although these features contain information on different modal data, they cannot fully reflect the three-dimensional geometric invariance of the object. Therefore, we propose a plug-and-play JDEM module which can extract the SDD for cross-modal data matching after inserting the JDEM module. The classification results can be obtained through a linear layer after extracting the SDD.

A diagram of the structure of the cross-dimensional data matching method is shown in Figure 3. Overall, our cross-dimensional matching method consists of two parts: the front-end traditional feature encoders and JDEM. The feature vector extracted by the front-end encoder and the random sampling point *P* are used as the input for the JDEM, and the SDD feature value satisfying the threshold requirement in the random sampling point *P* is used as an output to realize the mapping from a one-dimensional feature vector to a three-dimensional space descriptor. In the training process, it is necessary to input the sampling point *P* randomly. On the one hand, this can reduce the demand for memory; on the other hand, it can improve the network's generalization ability. In the training iteration process, the sampling point *P* is obtained by sampling the normalized coordinate points corresponding to the ground truth of the building SDF. The method compares the SDD value corresponding to the sampling point *P* obtained through the network's prediction with the ground truth and calculates the loss function to guide the network's training. It is worth noting that the SDD is a three-dimensional descriptor that can be visualized to facilitate the observation of the network training process and results.

Specifically, in order to verify the optimal encoder combination for our cross-dimensional matching method, we used the ResNet and PointNet series as the front-end image and point cloud encoder, respectively. ResNet18/34/50/101/152 refers to different network depths. The difference between PointNet and PointNet-k is whether or not the Knn module is added. After adding the Knn module, the point cloud will be searched for the nearest *k* points in each point space before entering the encoder. Then, these *k* + 1 points are merged and input into the encoder to improve the regional generalization of the encoder. Inspired by OccNet [44], we designed several different JDEM modules. The whole encoder combination is shown in Table 1, and the structures of the different JDEM modules are shown in Figure 4.

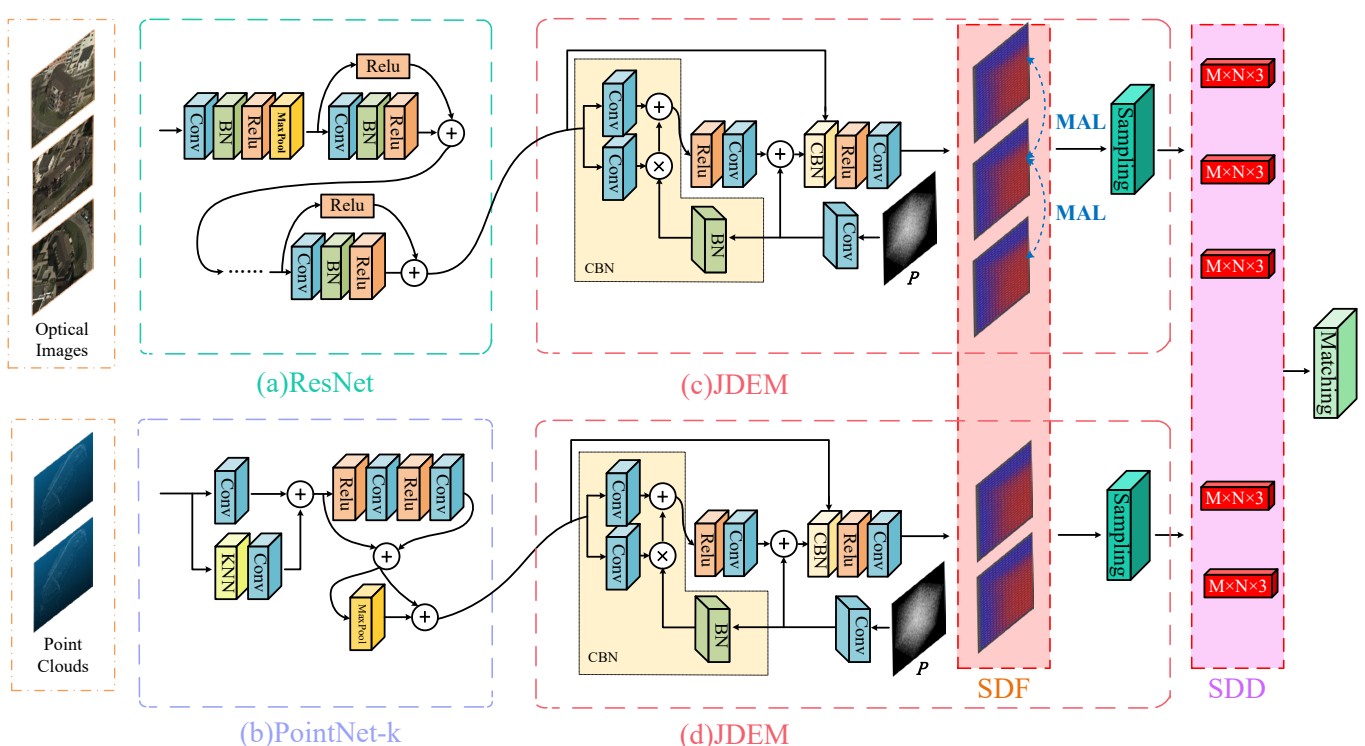

**Figure 3.** Structural diagram of the cross-dimensional data matching method. *P* denotes the random sampling points. In the data input process, it is necessary to input images from different perspectives of the same building, using Multi-View Adaptive Loss (MAL) constraint training at the same time.

**Table 1.** Various encoder combinations.

| Image Encoder | | Point Cloud Encoder | |
|---|---|---|---|
| **Part A** | **Part B** | **Part A** | **Part B** |
| ResNet18 | JDEM-CBN-RES | PointNet | JDEM-CBN-RES |
| ResNet34 | JDEM-CBN-None | PointNet-k | JDEM-CBN-None |
| ResNet50 | JDEM-BN-RES | | JDEM-BN-RES |
| ResNet101 | JDEM-None-RES | | JDEM-None-RES |
| ResNet152 | | | |

Joint Descriptor Extraction Module (JDEM).

### 2.3. Loss Function

The fusion of multiple loss functions is used to calculate our loss function. The overall objective function $L$ is a multi-task objective function divided into multistage weight loss and MAL. Multistage weight loss was proposed by the authors of [45].

#### 2.3.1. Multistage Weight Loss

Multistage weight loss is a function that calculates the difference between the SDD extracted from the image or point cloud and the ground truth. Ground truth is the SDF obtained by calculating the distance between the point and the model boundary in which the point is sampled from the unit space, and the model is the normalized 3D model of the buildings.

Compared with the typical method for calculating feature loss, which treats each position in the metric space uniformly, as for the L1 norm showed in (2), multistage weight loss focuses more on the selection of the boundary points on the model surface and gives greater attention to the points near the model surface. In contrast, the points far from the model surface are appropriately ignored, as shown in (3). In this way, the network's learning ability for building structures can be enhanced, and the SDD extraction effect can

be optimized. At the same time, although points far away from the model's surface receive less attention than others, they are less challenging to learn; thus, there is no negative impact on the results.

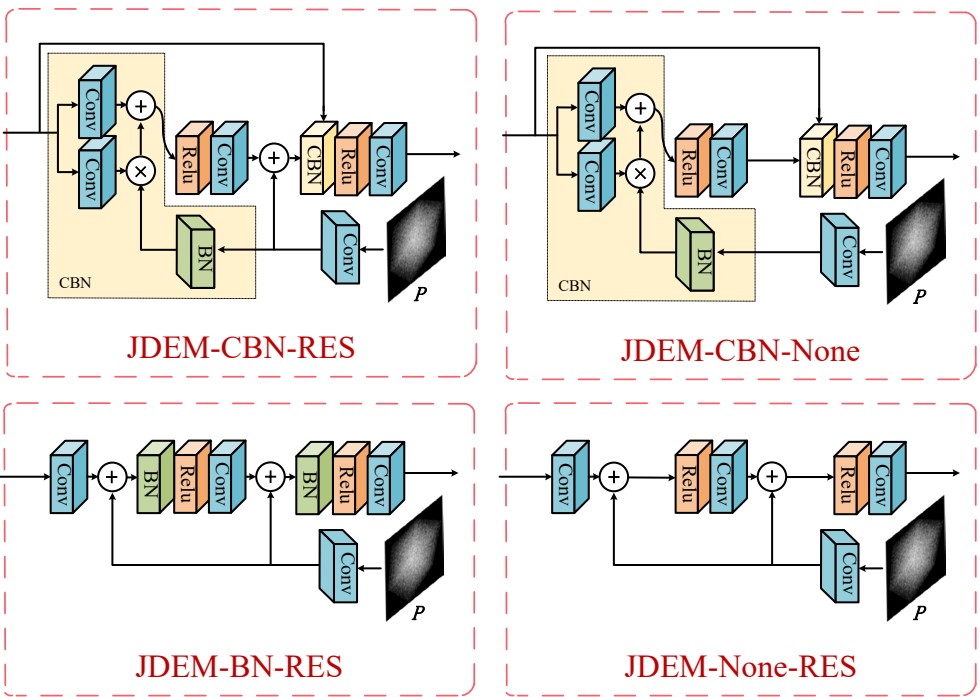

**Figure 4.** Structural diagram of different JDEMs.

$$L(F_\beta(I), GT) = \sum_{n=1}^{N} \left| F_\beta(I_n) - GT_n \right| \tag{2}$$

where $I_n$ is the image, $F_\beta$ is the SDF feature extraction network, $GT$ is ground truth, $N$ is the number of images.

$$L(F_\beta(I), GT) = \frac{1}{B} \sum_{b=1}^{B} \sum_{n=1}^{N} w(GT_{nb}) \times \left| F_\beta(I_n) - GT_n \right|_b \tag{3}$$

where $I_n$ is the image; $F_\beta$ is the SDF feature extraction network; $GT$ is ground truth; $N$ is the number of images; $B$ is the number of sampling points; $w(GT_{nb})$ is the weight corresponding to the current sampling point, determined by $GT_{nb}$; and $GT_{nb}$ is the SDF ground truth of the current sampling point, that is, the distance between the current point and the surface of the model. $w(GT_{nb})$ is shown in (4):

$$w(GT_{nb}) = \begin{cases} 2, & GT_{nb} \leq -0.02 \\ 4, & -0.02 < GT_{nb} < 0.02 \\ 1, & GT_{nb} \geq 0.02 \end{cases} \tag{4}$$

where $GT_{nb}$ is the SDF ground truth of the current sampling point.

Through the new feature loss function, the difference between the predicted value and the ground truth of the critical points can be amplified so that the feature extraction network can extract more accurate features through a more authentic relationship.

### 2.3.2. MAL

The same building is usually displayed in multiple remote-sensing images taken from different angles. Due to the difference in perspective, the buildings' features extracted from different images are usually different. Therefore, the concept of MAL is proposed to reduce

the feature distance of the images of the same building taken from different angles in the feature domain. MAL can improve the adaptability of the network to images of the same building taken from different angles and enhance the robustness of the features. Specifically, the image data, the building SDF ground truth, and the building ID are input into the network simultaneously during training. The distance of the SDD extracted from the image of the same building ID in the feature space is calculated, which alleviates the problem of cross-dimensional object-level matching errors caused by perspective differences. The calculation method of MAL is shown in (5) and (6):

$$L_{cls}(I^a, I^b) = \frac{1}{N} \sum_{n=1}^{N} \omega_{epoch} \times \exp(q \times D(I^a, I^b) + m) \tag{5}$$

$$D(I^a, I^b) = \sqrt{\sum_{n=1}^{N} (F_\beta(I^a)_n - F_\beta(I^b)_n)^2} \tag{6}$$

where $I^a$ and $I^b$ are image data of the same building ID, $N$ is the number of the building ID, $w$ and $q$ are weights, $m$ is offset, $D$ is the SDD feature distance of images taken from different angles, and $F_\beta$ is the SDD feature extraction network.

SDD is the mapping of the 3D structure of the building in the real physical world, but there are many building structures that are similar or even the same. Therefore, MAL only focuses on the distance of the SDDs extracted from the images of the same building ID and does not pay attention to others. The formula shown in (7) is used to load the data so as to improve the probability of different-angle images of the same building appearing in the same batch and not affecting the network's learning ability:

$$data\_index = \{a, a + n_a, a - n_a, \cdots, b, b + n_b, b - n_b\}, a \in \{0, S\} \tag{7}$$

where $S$ is the number of remaining samples in the current epoch, $a$ and $b$ are two random numbers from 0 to $S$, and $n_a$ and $n_b$ are the numbers of images of the current building taken from different angles in the current epoch.

Since the role of MAL is to shorten the distance of images of the same building taken from different angles in the descriptor space, MAL is not used at the beginning of the training. After the original loss reaches the best fit, MAL is added for training until the new best fit point is reached.

### 2.4. Proposed 2D-3D-CDOBM Data Set

In this section, the method for obtaining our data set is introduced first. The software named Labelme (v4.5.13) [46] is used to label the buildings in the airborne optical image on the object level, cut the object image according to the label, and resize it to 224 × 224 pixels, as shown in Figure 5a. The software named CloudCompare (v2.11.3) [47] is used to cut the LiDAR point clouds in the object-level space, as shown in Figure 5c. The software named Meshlab (v3.3) [48] is used to segment the 3D models corresponding to the optical images and LiDAR point clouds in order to obtain the object-level model of a single building, as shown in Figure 5b.

In order to verify the effect of our object-level joint descriptor in cross-dimensional matching tasks, an object-level data set composed of building images and corresponding point clouds is needed. In addition, to learn how to extract the SDD accurately, a building model corresponding to cross-dimensional data is necessary. Therefore, a data set consisting of object-level optical images, object-level point clouds, and 3D models of the corresponding buildings is produced. The numbers of various types of data points in the data set are shown in Table 2.

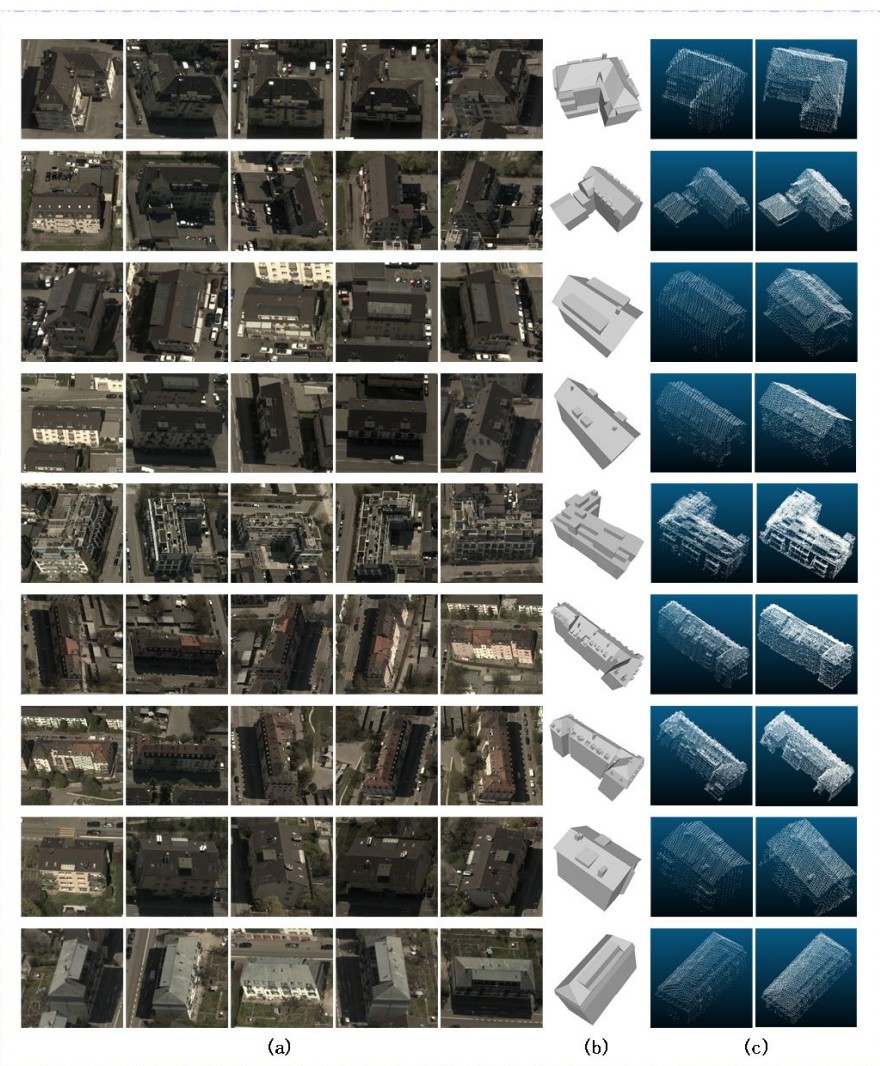

**Figure 5.** A schematic diagram of the 2D-3D-CDOBM data set. (**a**) is 2D optical image, (**b**) is building model, and (**c**) is 3D point cloud.

**Table 2.** Details of our data set.

| Object-Level Optical Image | Object-Level Point Cloud | Object-Level 3D Model |
| --- | --- | --- |
| 11,094 | 906 | 453 |

### 2.4.1. Optical Images

The Institute of Geodesy and Photogrammetry, ETH Zurich, provides airborne optical images of the Zurich region that are published in the ISPRS data set [49]. These optical images cover an area nearby the center of Zurich (Switzerland) and the ETH Hoenggerberg. The region comprises the center of the quarter of Hoengg, including different residential areas with different types of buildings (flat roofs, hip roofs, etc.)

The Zurich Hoengg data set is based on aerial photographs collected over Zurich in 1995. The data set consists of four aerial images of the city of Zurich taken at an average image scale of ca. 1:5000. The 23 cm × 23 cm color photographs were scanned at 14 μm, yielding color images of approximately 840 Mb. Each image is approximately 16,500 × 16,400 pixels. The photography equipment was flown ca. 1050 above the ground with 65% forward and 45% sideward overlap. The camera was a Leica RC20. We annotated, segmented, and produced a data set containing only 1 building per image, a total of 11,094 images of 453 houses.

### 2.4.2. Point Clouds

Twice in 2014 and again from 2017 to 2018, the Federal Government conducted two high-resolution laser scans of the geographical area of the canton of Zurich (E/N Min:2669255/1223895; E/N Max:2716900/1283336[m]) using the airborne LiDAR system method (minimum point density of 5 Pkt/m$^2$). The reference system, CH1903+_LV95, has a position accuracy of $\pm 0.2$ (m) and a height accuracy of $\pm 0.1$ (m). The Swiss SURFACE 3D product and spatial data set (GIS-ZH No.24) were created. We segmented the object-level buildings and saved individual buildings as point cloud files, yielding a total of 906 point cloud files of 453 buildings.

### 2.4.3. Three-Dimensional Model and SDF Ground Truth

The CityGML data of Zurich is available at "https://3D.bk.tudelft.nl/opendata/ (accessed on 24 January 2022)", which includes the 3D models of buildings. After object-level splitting, SDF is extracted as the ground truth and input into the training network.

### 2.5. Evaluation Metrics

After extracting the SDD, PointNet [42] is used to classify the descriptors for cross-dimensional matching. In order to verify the precision of the SDD in presenting the 3D geometric information of the object, the EMD [50] is used to calculate the similarity between the SDD extracted from the 2D image and the SDD extracted from the 3D point cloud. EMD can intuitively describe the distance of the SDDs in the descriptor space. The smaller the EMD is, the more similar the SDDs are. Moreover, to verify the matching effect of our method, instance accuracy and category accuracy are used as metrics to evaluate the matching accuracy of the total sample and matching category accuracy of the building, respectively.

### 2.5.1. Instance Accuracy

Instance accuracy refers to the percentage of correctly matched data in the total data to be matched, as shown in (8):

$$Ins\_acc = \frac{TP}{TP + FP} \times 100\% \tag{8}$$

where TP is the correct matching sample, and FP is the wrong matching sample.

### 2.5.2. Category Accuracy

Each building object is regarded as a class, and the category accuracy is the average accuracy of all the categories, as shown in (9):

$$Cat\_acc = \frac{1}{N} \sum_{n=1}^{N} A_n \times 100\% \tag{9}$$

where $A_n$ is the accuracy of each category, and $N$ is the total number of categories.

### 2.6. The Training Platform and Parameter Settings

All experiments were carried out in the same environment. Training and testing of the network were conducted under Ubuntu 18.04. The hardware environment included a single Intel i7 9700 CPU, and the GPU acceleration used a single NVIDIA RTX 2080 Super with 8 GB of memory. In training, the batch size was 64, the Adam optimizer is used, the learning rate was set to $1 \times 10^{-4}$, and the epoch was 1500. We conducted multiple sets of experiments under different hyperparameter setting conditions and judged the optimal hyperparameter settings according to the experimental results. The number of points sampled from the training models was 100,000, with a distance in the range of $-0.3$ to $0.3$ (based on the normalized model).

In these experiments, the 2D images were divided into training, validation, and test data in a ratio of 6:2:2. The 3D point clouds scanned in 2014 were used for training and validation, and the 3D point clouds scanned in 2017–2018 were used for testing.

## 3. Experiment and Results

In this section, we verify the effect of SDD extraction and the accuracy of cross-dimensional matching, respectively, under different encoder combinations based on a large number of ablation experiments. We evaluate the accuracy of different encoder combinations' SDDs by calculating the distance between descriptors in the descriptor space and evaluate its influence on the accuracy of cross-dimensional matching according to the sample accuracy and category accuracy. Specifically, the EMD is a description of the descriptor quality. The smaller the EMD is, the better the descriptor's description of the target's three-dimensional information will be. While the instance accuracy is the accuracy of all the data to be matched, the category accuracy is the average accuracy of all the categories (building ID). These are representations of the descriptor-matching ability.

### 3.1. Extraction Effects of Different Descriptors for Cross-Dimensional Matching
3.1.1. Extraction Effect under Different Encoder Combinations

The control variable method is used to intuitively obtain the influences of different encoder combinations on the matching results. Specifically, when performing the image-point-cloud-matching task, the image descriptors are extracted by the encoder combination "ResNet18 + JDEM-CBN-RES" and matched with the point cloud descriptors extracted by various encoder combinations. The same goes for the point-cloud-matching images. When performing the task of image point cloud matching, the descriptor of the point cloud is extracted by the encoder combination "PointNet + JDEM-CBN-RES". In addition, since our EMD calculation method does not normalize the SDD in advance, the range of the results is not fixed, and its value is related to the size of the building model. Firstly, the superiority of the SDD over the SOPD is verified. The results are shown in Tables 3 and 4.

**Table 3.** The EMD between descriptors extracted from 2D images and 3D point clouds under different encoder combinations.

| Image Encoder | | SOPD | SDD |
|---|---|---|---|
| Part A | Part B | | |
| ResNet18 | JDEM-CBN-RES | 74.6084 | **<u>42.7909</u>** (−31.8175) |
| ResNet34 | JDEM-CBN-RES | 75.7076 | 128.6485 (+52.9409) |
| ResNet50 | JDEM-CBN-RES | 77.0661 | 63.7674 (−13.2987) |
| ResNet101 | JDEM-CBN-RES | **<u>73.8090</u>** | 45.9396 (−27.8694) |
| ResNet152 | JDEM-CBN-RES | 76.7682 | 46.1051 (−30.6631) |
| ResNet18 | JDEM-BN-RES | 77.9651 | 43.2792 (−34.6859) |
| ResNet18 | JDEM-None-RES | 77.7802 | 43.2648 (−34.5154) |
| ResNet18 | JDEM-CBN-None | 74.4358 | 43.1497 (−31.2861) |

In the table, the bold underline represents the best result among the comparison algorithms in the same column, and the content in parentheses shows the difference in evaluation indicators between the same row. If the Evaluation Metrics becomes smaller, it is displayed in green; if it becomes larger, it is displayed in red. Specifically, for the EMD, smaller values are better, while for the accuracy, larger values are better. The following tables are the same. Signed Distance Descriptor (SDD).

**Table 4.** The EMD between descriptors extracted from 3D point clouds and 2D images under different encoder combinations.

| Point Cloud Encoder | | SOPD | SDD |
|---|---|---|---|
| Part A | Part B | | |
| PointNet | JDEM-CBN-RES | **<u>74.6084</u>** | **<u>42.7909</u>** (−31.8175) |
| PointNet | JDEM-BN-RES | 77.9872 | 46.2336 (−31.7536) |
| PointNet | JDEM-CBN-None | 75.4318 | 42.9513 (−32.4805) |
| PointNet | JDEM-None-RES | 76.3757 | 46.9127 (−29.4630) |

From the above experimental results, it can be seen that the distance in the descriptor space between the extracted SDD and ground truth is closer than that of the SOPD. In other words, the SDD describes the 3D geometric invariance of the object more accurately. The following section verifies the influence of the proposed MAL on the SDD extraction effect.

3.1.2. Extraction Effect under Different Proposed Methods

In this experiment, the number in "JDEM-CBN-RES+MAL-0%" refers to the ratio of MAL to the original function multistage weight loss. The results are shown in Table 5.

**Table 5.** The EMD between descriptors extracted from 2D images matching 3D point clouds using different ratios of MAL.

| Image Encoder | | SDD |
|---|---|---|
| Part A | Part B | |
| ResNet18 | JDEM-CBN-RES+MAL-0% | 42.7909 |
| ResNet18 | JDEM-CBN-RES+MAL-5% | 42.1315 ($-0.6594$) |
| ResNet18 | JDEM-CBN-RES+MAL-10% | 41.9423 ($-0.8486$) |
| ResNet18 | JDEM-CBN-RES+MAL-20% | **41.5696** ($-1.2213$) |
| ResNet18 | JDEM-CBN-RES+MAL-30% | 42.3517 ($-0.4392$) |
| ResNet18 | JDEM-CBN-RES+MAL-40% | 41.7006 ($-1.0903$) |
| ResNet18 | JDEM-CBN-RES+MAL-50% | 42.1304 ($-0.6605$) |
| ResNet18 | JDEM-CBN-RES+MAL-60% | 42.6350 ($-0.1559$) |
| ResNet18 | JDEM-CBN-RES+MAL-70% | 41.9715 ($-0.8194$) |
| ResNet18 | JDEM-CBN-RES+MAL-80% | 42.0579 ($-0.7330$) |
| ResNet18 | JDEM-CBN-RES+MAL-90% | 42.1455 ($-0.6454$) |
| ResNet18 | JDEM-CBN-RES+MAL-100% | 41.5711 ($-1.2198$) |

From the experimental results, it can be seen that the addition of MAL improves the extraction effect of the SDD. Meanwhile, the setting of different weights for MAL has no obvious causal relationship.

Finally, we identified the value of the $k$ of PointNet-k that is most suitable for our proposed data set. The results are shown in Table 6.

**Table 6.** The EMD between descriptors extracted from 3D point clouds matching 2D images with different $k$ of PointNet-k.

| Point Cloud Encoder | | SDD |
|---|---|---|
| Part A | Part B | |
| PointNet | JDEM-CBN-RES | 42.7909 |
| PointNet-2 | JDEM-CBN-RES | 44.4586 (+1.6677) |
| PointNet-4 | JDEM-CBN-RES | 41.9119 ($-0.8790$) |
| PointNet-6 | JDEM-CBN-RES | 42.9444 (+0.1535) |
| PointNet-8 | JDEM-CBN-RES | **41.1428** ($-1.6481$) |
| PointNet-10 | JDEM-CBN-RES | 41.7385 ($-1.0524$) |
| PointNet-12 | JDEM-CBN-RES | 42.2998 ($-0.4911$) |

According to the above experimental results, it can be seen that the value of $k$ influences the extraction effect of the SDD. Because the value of $k$ determines the spatial coding range of PointNet-k, when $k$ is small, the nearest neighbor points are too few in number to support PointNet-k in learning. This is why the quality of the SDD extracted by "PointNet-2 + JDEM-CBN-RES" and "PointNet-6 + JDEM-CBN-RES" does not improve. In conclusion, PointNet-k works best when $k$ is 8. The change curve of the SDD distance with different ratios of MAL and different parameters $k$ of PointNet-k is shown in Figure 6.

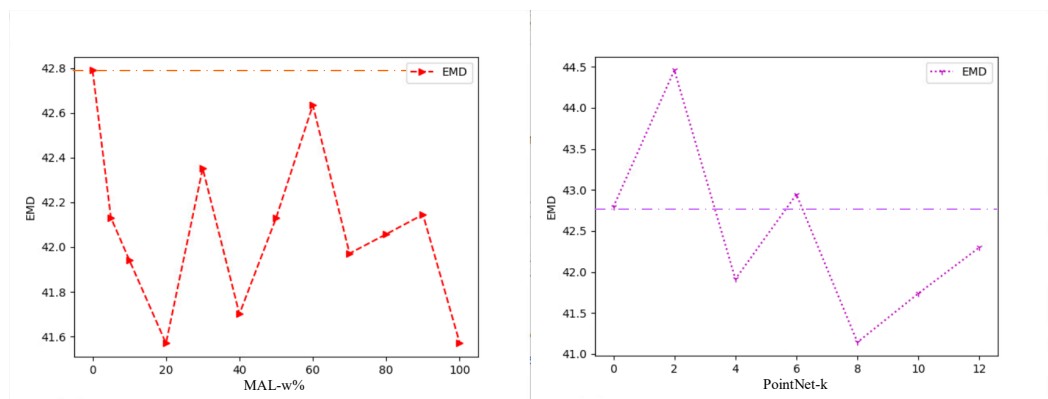

**Figure 6.** The change curve of the SDD distance with different ratios of MAL and different values of parameter *k* of PointNet-k.

### 3.2. Cross-Dimensional Matching Accuracy under Different Encoder Combinations

In order to accurately evaluate the effectiveness of different encoder combinations and the proposed method for cross-dimensional matching tasks, we continued to use the sample accuracy and category accuracy to evaluate the cross-dimensional matching results. The experiment reported in this chapter also used the control variable method, and the default condition settings were the same as those in Section 3.1. In this experiment, the comparison between the best matching results based on the SDD and the SOPD matching results, which were divided into 2D images matching 3D point clouds and 3D point clouds matching 2D images, was conducted first. The results are shown in Tables 7 and 8.

**Table 7.** The best accuracy for 2D images matching 3D point clouds.

| SOPD | | SDD | |
|---|---|---|---|
| **Instance acc** | **Category acc** | **Instance acc** | **Category acc** |
| 72.2582% | 69.6282% | **76.8301%** (+4.5719%) | **73.2332%** (+3.605%) |

**Table 8.** The best accuracy for 3D point clouds matching 2D images.

| SOPD | | SDD | |
|---|---|---|---|
| **Instance acc** | **Category acc** | **Instance acc** | **Category acc** |
| 75.2193% | 74.9485% | **79.8559%** (+4.6366%) | **79.6946%** (+4.7461%) |

According to the experimental results, it can be seen that the instance accuracy is generally higher than the category accuracy, which indicates that there is an imbalance in the matching ability between categories. In other words, the matching accuracy of some buildings is significantly higher than that of other buildings. Furthermore, the experimental results show that our cross-dimensional matching method is superior to the baseline in terms of both instance accuracy and category accuracy.

Next, the ablation experiment was divided into three parts. The first part verified the superiority of the SDD compared with the SOPD. The results are shown in Tables 9 and 10. The second part verified the effectiveness of MAL, and the results are shown in Table 11. Finally, the value of the encoder *k* that is most suitable for our proposed data set was identified, and the results are shown in Table 12.

**Table 9.** The accuracy of 2D images matching 3D point clouds under different encoder combinations.

| Image Encoder | | SOPD | | SDD | |
|---|---|---|---|---|---|
| Part A | Part B | Instance acc | Category acc | Instance acc | Category acc |
| ResNet18 | JDEM-CBN-RES | 72.2582% | 69.6282% | 75.5470% (+3.2888%) | 72.4163% (+2.7881%) |
| ResNet34 | JDEM-CBN-RES | 72.0759% | 69.4502% | 71.7247% (−0.3512%) | 68.8125% (−0.6377%) |
| ResNet50 | JDEM-CBN-RES | 72.0556% | 69.4846% | 74.5746% (+2.5190%) | 71.5551% (+2.0705%) |
| ResNet101 | JDEM-CBN-RES | **73.4063%** | **71.1805%** | **76.5532%** (+3.1469%) | **73.3839%** (+2.2034%) |
| ResNet152 | JDEM-CBN-RES | 71.8395% | 68.8852% | 75.1351% (+3.2956%) | 72.2858% (+3.4006%) |
| ResNet18 | JDEM-BN-RES | 66.8828% | 63.0065% | 73.0821% (+6.1993%) | 68.3356% (+5.3291%) |
| ResNet18 | JDEM-None-RES | 68.4292% | 65.3175% | 73.3793% (+4.9501%) | 69.7284% (+4.4109%) |
| ResNet18 | JDEM-CBN-None | 71.1642% | 68.3775% | 76.2763% (+5.1121%) | 72.5993% (+4.2218%) |

**Table 10.** The accuracy of 3D point clouds matching 2D images under different encoder combinations.

| Image Encoder | | SOPD | | SDD | |
|---|---|---|---|---|---|
| Part A | Part B | Instance acc | Category acc | Instance acc | Category acc |
| PointNet | JDEM-CBN-RES | 71.8985% | 71.6691% | 72.3997% (+0.5012%) | 72.2247% (+0.5556%) |
| PointNet | JDEM-BN-RES | 75.2193% | 74.9485% | **76.4098%** (+1.1905%) | **76.2344%** (+1.2859%) |
| PointNet | JDEM-None-RES | 71.2719% | 71.3812% | 73.8722% (+2.6003%) | 73.6228% (+2.2416%) |
| PointNet | JDEM-CBN-None | 71.2719% | 71.3812% | 73.8722% (+2.6003%) | 73.6228% (+2.2416%) |

**Table 11.** The accuracy of 2D images matching 3D point clouds using different ratios of MAL.

| Image Encoder | | SDD | |
|---|---|---|---|
| Part A | Part B | Instance acc | Category acc |
| ResNet18 | JDEM-CBN-RES+MAL-0% | 75.5470% | 72.4163% |
| ResNet18 | JDEM-CBN-RES+MAL-5% | 76.1750% (+0.6208%) | 72.6685% (+0.2522%) |
| ResNet18 | JDEM-CBN-RES+MAL-10% | 76.3979% (+0.8509%) | 72.6124% (+0.1961%) |
| ResNet18 | JDEM-CBN-RES+MAL-20% | 75.9184% (+0.3714%) | 72.6139% (+0.1976%) |
| ResNet18 | JDEM-CBN-RES+MAL-30% | 76.0940% (+0.5470%) | 72.4914% (+0.0751%) |
| ResNet18 | JDEM-CBN-RES+MAL-40% | 76.2966% (+0.7496%) | **73.4532%** (+1.0369%) |
| ResNet18 | JDEM-CBN-RES+MAL-50% | 76.3033% (+0.7563%) | 72.9010% (+0.4847%) |
| ResNet18 | JDEM-CBN-RES+MAL-60% | 76.3911% (+0.8441%) | 73.1598% (+0.7435%) |
| ResNet18 | JDEM-CBN-RES+MAL-70% | 75.7361% (+0.1891%) | 71.9418% (−0.4745%) |
| ResNet18 | JDEM-CBN-RES+MAL-80% | 76.6140% (+1.0670%) | 73.0335% (+0.6172%) |
| ResNet18 | JDEM-CBN-RES+MAL-90% | 76.7896% (+1.2426%) | 73.4325% (+1.0162%) |
| ResNet18 | JDEM-CBN-RES+MAL-100% | **76.8301%** (+1.2831%) | 73.2332% (+0.8169%) |

**Table 12.** The accuracy of 3D point clouds matching 2D images using different k of PointNet-k.

| Point Cloud Encoder | | SDD | |
|---|---|---|---|
| Part A | Part B | Instance acc | Category acc |
| PointNet | JDEM-CBN-RES | 76.4098% | 76.2344% |
| PointNet-2 | JDEM-CBN-RES | 73.1830% (−3.2268%) | 73.0390% (−3.1954%) |
| PointNet-4 | JDEM-CBN-RES | 76.9110% (+0.5012%) | 76.6826% (+0.4482%) |
| PointNet-6 | JDEM-CBN-RES | 77.8822% (+1.4724%) | 77.6392% (+1.4048%) |
| PointNet-8 | JDEM-CBN-RES | **79.8559%** (+3.4461%) | **79.6946%** (+3.4602%) |
| PointNet-10 | JDEM-CBN-RES | 78.3835% (+1.9737%) | 78.2929% (+2.0585%) |
| PointNet-12 | JDEM-CBN-RES | 76.3784% (−0.0314%) | 76.1185% (−0.1159%) |

Through the above experimental results, the same conclusion as that outlined Section 3.1 can be drawn: a higher matching accuracy can be obtained by extracting the SDD, as compared with the SOPD; MAL improves the matching accuracy; and the weight of MAL does not affect its functionality. Since the same category contains multiple building images taken from different angles, the quality of descriptors extracted from building images taken from different angles and the instance accuracy and category accuracy of

cross-dimensional matching are improved after using MAL. Therefore, it can be proved that MAL improves the adaptability of the image encoder module to images taken from different angles.

It is evident in Figure 7 that with the increase in $k$, the matching accuracy of PointNet-k undergoes a process of decline–rise–decline. The SDD distance curve in Figure 7 can provide a similar result to that in Figure 6. This is because the value of $k$ determines the spatial coding range of PointNet-k. When $k$ is small, the nearest neighbor point is too small to support PointNet-k learning. When $k$ is too large, the wrong nearest neighbor causes PointNet-k to learn the wrong spatial encoding. The point cloud density of the data determines the optimal value of $k$. In conclusion, it can be found that PointNet-k helps to improve the accuracy of cross-dimensional matching. For our data set, the best matching result can be obtained when $k = 8$.

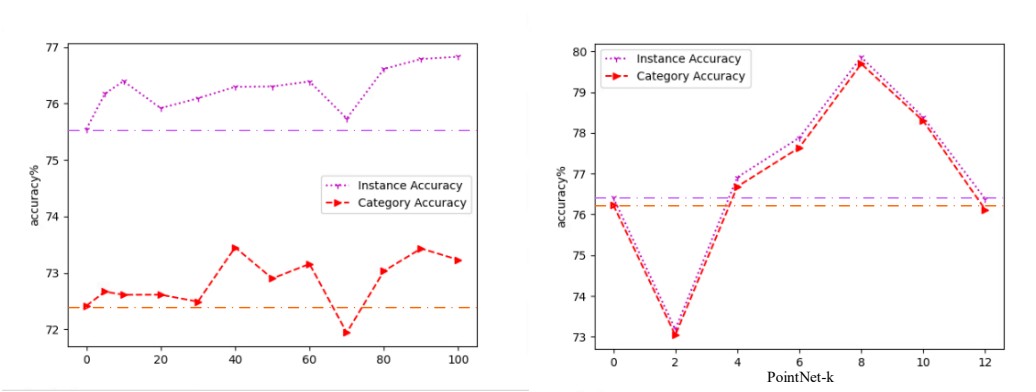

**Figure 7.** The change curve of cross-dimensional matching accuracy with different ratios of MAL and different parameters $k$ of PointNet-k.

## 4. Discussion

This section can be divided into two parts. The first part provides a discussion to further explore the specific advantages of our approach, the need for further research, and the potential applications of this research. The second part provides a discussion to explain the previous experimental results, which includes further elaboration of the experimental results, an analysis of the impact on the experimental data set, and future research directions.

Our method is a new method called JDEM which aims to extract a joint descriptor named SDD from cross-dimensional data. Through this method, the inherent modal differences between 2D images and 3D point cloud data are solved, and object-level matching relationships can accurately be established between different dimensional data. This research is of great practical value. For example, in the case of a UAV losing its control signal and positioning signal due to targeted interference, the use of the data matching method can help the UAV to complete self-positioning so that it can continue to complete the established task independently under offline conditions. However, due to the wide range of shooting angles and the great changes in light conditions in traditional image matching tasks, the image–image matching positioning method requires the storage of a large number of images of the UAV as a database. With the increase in the number of 2D images taken from different angles and lighting conditions in the same area in the database, the accuracy of positioning will also increase, but this will consume a lot of terminal storage space. The use of 3D point cloud data can solve this problem. Firstly, because the 3D point cloud data composed of coordinates do not have shooting angle and light condition problems, the image-point-cloud-matching method requires less storage space at the terminal than the image-matching positioning method. Secondly, the image-point-cloud-matching method is not affected by the quality or number of images in the

database, which reduces the possibility of error matching caused by the data imbalance problem in the database. Therefore, our proposed method has good application value.

In this study, our method was compared with other methods for object-level cross-dimensional matching. Despite the better results achieved with our network, several issues remain to be addressed. Firstly, the joint descriptor proposed for our method is based on the three-dimensional features of the structure and does not contain the color information of the building. Therefore, it is easy to mismatch the faces of similar structures or the same building structure with different colors. There are many ways to obtain point clouds with color texture information [51]. If the color information in such point cloud data can be added to the descriptor extraction process, the above problems can effectively be alleviated. Moreover, a problem that must be addressed is the data imbalance. Due to the characteristics of deep learning, structures with more occurrences in the training set can achieve better learning results. In comparison, structures with lower occurrences in the data set are difficult to understand and accurately learn. The data of just over 400 buildings are not enough to fully support the generalization learning of the network. Therefore, more buildings are necessary for the data set.

## 5. Conclusions

This paper proposed a new object-level cross-dimensional building-matching method based on the SDD. The SDD is a descriptor extracted by the JDEM which uses the 3D geometric invariance of buildings to achieve the cross-dimensional matching of buildings. In addition, MAL was designed to improve the adaptability of the image encoder module for images taken at different angles. A 2D-3D-CDOBM data set was proposed to verify the effectiveness and robustness of our method. A large number of experimental results showed that our object-level descriptor extraction method achieves state-of-the-art outcomes. The effectiveness of our proposed module was also proven. Although this method marks apparent progress compared with the baseline, SDD can still only describe the 3D geometry of the object and needs a better resolution when handling different buildings with similar shapes. This is a challenging issue. Scenario-level cross-dimensional data matching may be an effective solution, which is the critical issue that we are committed to addressing.

**Author Contributions:** Conceptualization and funding acquisition, C.Z., Y.Y. and W.H.; methodology and writing of the original draft, W.W. and Y.Y.; processing and analysis of cross-dimensional data, visualization, and editing of the manuscript, N.S. and S.F.; ablation experiment and revision of the manuscript, Q.X. All authors have read and agreed to the published version of the manuscript.

**Funding:** This research was funded by the National Natural Science Foundation of China (No. 62271159, No. 62071136, No. 62002083, No. 61971153); Heilongjiang Outstanding Youth Foundation (YQ2022F002); Heilongjiang Postdoctoral Foundation (LBH-Q20085 and LBH-Z20051); Fundamental Research Funds for the Central Universities Grant (3072022QBZ0805, 3072021CFT0801, and 3072022CF0808; and High-Resolution Earth Observation Major Project (grant No. 72-Y50G11-9001-22/23).

**Data Availability Statement:** The proposed data set will be available to download at 'https://github.com/HEU-super-generalized-remote-sensing/Cross-dimensional-Object-level-Matching-Data-Set.git (accessed on 4 June 2023)'. For the code, please contact the corresponding author via email.

**Acknowledgments:** The authors express their thanks for the provision of the data sets by ISPRS and EuroSDR, released in conjunction with the ISPRS scientific initiative 2014 and 2015, led by ISPRS ICWG I/Vb.

**Conflicts of Interest:** The authors declare no conflict of interest.

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
