# Peer review of "A Novel Object-Level Building-Matching Method across 2D Images and 3D Point Clouds Based on the Signed Distance Descriptor (SDD)"

_remotesensing, doi:10.3390/rs15122974_

Round 1
Reviewer 1 Report
Overall, the manuscript provides a comprehensive and well-structured study on a novel object-level buildings matching method across cross-dimensional data. The study is interesting and of practical values.
I recommend providing more detailed information on the specific benefits or applications of the proposed method. Consider expanding the discussion to highlight potential implications or advantages that arise from the use of the proposed approach. This addition would provide readers with a clearer context for the significance of the work and its potential impact in relevant domains.
- The study introduces a novel method called JDEM, which aims to enhance descriptor extraction for building matching across imagery and LiDAR data. The method is both intriguing and thoroughly explained.
- Further discussion is needed to address the existing research gaps. Similar to my previous comment, the author should begin by highlighting the specific advantages of the proposed method, such as the rationale behind matching buildings using cross-dimensional data. While I believe lines 43-57 touch upon this matter, this paragraph appears to be somewhat confusing. It seems the author intends to convey that deep learning models rely on a substantial amount of high-quality data. Is the author suggesting that the quality of open-source imagery data varies significantly? Additionally, what are the respective benefits of using imagery and LiDAR?
- Although the introduction is comprehensive, it fails to explicitly address the challenges associated with correlating imagery and LiDAR descriptors.
- The discussion section lacks sufficient depth. In line with my previous comment, the author should elaborate more on how the proposed method contributes to the existing knowledge. Are there any practical applications or specific use cases that can be explored?
- I do not believe that a dataset alone can be considered a substantial contribution.
- The language used in the paper could be improved. There are several instances of wordy and unclear sentences that need revision.
The language is fine.
Author Response
要点 1: 我建议提供有关所提议方法的具体好处或应用的更详细信息。考虑扩大讨论范围,以突出使用拟议方法产生的潜在影响或优势。这一补充将为读者提供更清晰的背景,了解这项工作的重要性及其在相关领域的潜在影响。
响应 1:
首先,非常感谢您的评论。针对您的问题,我们概括了所提出的方法的具体优点和优势,并在引言的第95段102-<>行中添加了详细的可能应用领域和影响。以便读者更容易理解这项工作的重要性及其在相关领域的潜在影响。更具体的描述将在讨论部分提及。再次感谢您的评论。
要点 2:需要进一步讨论以解决现有的研究差距。与我之前的评论类似,作者应该首先强调所提出的方法的具体优点,例如使用跨维度数据匹配建筑物的基本原理。虽然我认为第43-57行涉及这个问题,但这一段似乎有些令人困惑。作者似乎打算传达深度学习模型依赖于大量高质量数据。作者是否暗示开源图像数据的质量差异很大?此外,使用影像和激光雷达分别有什么好处?
响应 2:
首先非常感谢您的评论。
我们想在第 43-57 行说的如下。通过图-图匹配进行无人机定位的方法是,当Query中待匹配的图片成功匹配图库中的某张图片时,将图库中图片的地理位置信息作为待匹配图片的信息。因此,无论是传统的图像特征提取方法还是深度学习模型方法,要通过图像-图像匹配实现定位,都需要在数据库中存储足够的具有目标位置信息的图像作为Gallery,以实现精确匹配。
正如我们在讨论部分添加的,与图像-图像匹配相比,跨维度匹配方法的优点如下。首先,二维图像的优点是容易获得,但要实现精确匹配需要大量的图像。虽然由坐标组成的三维点云数据不存在拍摄角度和光照条件问题,但图像点云匹配方法比图匹配定位方法在终端需要更少的存储空间。其次,图像-点云匹配方法不受数据库中图像质量和数量的影响,降低了数据库中数据不平衡问题导致错误匹配的可能性;
根据您的意见,我们修改了原来的第43-57行和讨论部分,再次感谢您的评论。
观点3:尽管引言很全面,但它未能明确解决与关联图像和LiDAR描述符相关的挑战。
响应 3:
首先非常感谢您的评论。
将2D图像与3D点云匹配的困难在于如何解决它们之间属性的固有差异,如论文第71-74行所述:“2D图像通常是场景外观的二维映射,而3D点云编码场景的结构。如何从不同维度的数据中提取可用于同一描述符空间中跨维匹配的描述符是关联影像和 LiDAR 描述符的相关挑战。根据您的意见,我们在介绍和讨论部分添加了相关表达。再次感谢。
要点4:讨论部分缺乏足够的深度。根据我之前的评论,作者应该更多地阐述所提出的方法如何有助于现有知识。是否有任何可以探索的实际应用或特定用例?
响应 4:
感谢您的评论。我们已根据您的意见对讨论部分进行了修改。
观点5:我不认为数据集本身可以被视为实质性贡献。
响应 5:
首先非常感谢您的评论。
由于创建数据集不涉及理论创新,因此数据集本身的贡献并不大。但是,我们已经创建了跨维对象级匹配领域的第一个数据集,该数据集完全由真实数据组成。它为以下研究提供了数据基础。我们认为这至少是一个很小的贡献,因此我们将此贡献作为贡献中的最后一个。再次感谢您的评论。
第6点:文件中使用的语言可以改进。有几个冗长和不清楚的句子需要修改。
响应 6:
非常感谢您的评论。我们再次仔细校对全文,修改语法错误和错别字,润色语言。我们还修改了表格以提高可读性和美观性。修改后的部分已在手稿中注明。感谢您的评论。

Reviewer 2 Report
In this paper, authors proposed an object-level buildings matching method across cross-dimensional data including 2D images and 3-D point clouds. The core of this method is proposing a plug-and-play Joint Descriptor Extraction Module (JDEM) to extract descriptors containing buildings’ three-dimensional shape information from object-level remote sensing data of different dimensions for matching. Authors did a good work and interested for the readers and the following review comments are recommended, and the authors are invited to explain and modify.
1 “The loss function improves the adaptability of the image encoder module to images from different angles”, Experimental justification for this sentence is not given.
2 “In this part, the method of making our data set will be introduced first”, authors did not give the references of software’s used to label.
3 “In training, the batch size is 64, the Adam optimizer is used, the learning rate is”, how to optimize hyperparameters?
4 Mention the limitations and future works of the developed system elaborately.
5 Authors should mention the implementation challenges.
6 The paper needs intensive proofreading as it contains many long, inconsistent sentences. Besides, the manuscript contains many grammar errors and typos.
1. The core of this method is proposing a plug-and-play Joint Descriptor Extraction Module (JDEM) to extract descriptors containing buildings’ three-dimensional shape information from object-level remote sensing data of different dimensions for matching. 2. Yes 3. A novel object-level building matching method across 2-D images and 3-D point clouds is studied. 4. Should mention the implementation challenges. 5. Need to justify, for example “The loss function improves the adaptability of the image encoder module to images from different angles”, Experimental justification for this sentence is not given. 6. Need more refs in introduction. 7. Improve qualities of tables.Extensive editing of English language required
Author Response
Point 1: “The loss function improves the adaptability of the image encoder module to images from different angles”, Experimental justification for this sentence is not given.
Response 1:
Thank you very much for your comments firstly.
In the experiment, we use three evaluation indicators to evaluate the effectiveness of our proposed method, which are descriptor space distance EMD, instance accuracy and category accuracy. Specifically, the EMD is a description of the descriptor quality, the instance accuracy is the accuracy of all the data to be matched, and the category accuracy is the average accuracy of all categories ( building ID ). Therefore, the decrease of EMD indicates the improvement of descriptor quality, and the instance accuracy and category accuracy are the representations of descriptor matching ability. Among them, the instance accuracy refers to the overall accuracy of the instance, and the category accuracy is a representation of whether the instance accuracy distribution in each category is balanced. For example, the instance accuracy is unchanged and the category accuracy is increased, which shows that the new method has better adaptability to different categories. The category accuracy remains unchanged and the instance accuracy increases, indicating that although the overall effect of the new method is improving, the adaptability to different categories is declining, and the category learning ability is more uneven.
In the ablation experiment of MAL, the instance accuracy and category accuracy are both increasing after adding MAL, indicating that the number of correctly matched instances is increasing, and the number of correct instances in each category is also increasing. Due to the angle difference between different images in each category, combined with the phenomenon that EMD is also decreasing, it can be concluded that after adding MAL, the descriptor extraction effect and matching accuracy of the proposed method for different angle images are improved, that is, the adaptability to different angle images is improved.
From another perspective, judging whether MAL improves the adaptability of the image encoder module to images from different angles is to see whether the descriptors extracted from images from different angles are more accurate, and whether the accuracy of cross-dimensional matching of images from different angles is higher. The experimental results obviously prove this, so it can also be concluded that MAL improves the adaptability of the image encoder module to images from different angles.
In response to your suggestions, we added an explanation of the experimental results in lines 464-468 of the article, and thank you again for your comments.
Point 2: "In this part, the method of making our data set will be introduced first", authors did not give the references of software’s used to label.
Response 2:
Thank you very much for your comments. We have added three references of related software.
Point 3: “In training, the batch size is 64, the Adam optimizer is used, the learning rate is”, how to optimize hyperparameters?
Response 3:
Thank you very much for your comments firstly.
We conduct multiple sets of experiments under different hyperparameters setting conditions and judge the optimal hyperparameter setting according to the experimental results, and optimize the hyperparameter setting in this way. According to your suggestion, we add an introduction to the hyperparameter optimization method in line 354-356, and thank you again for your comments.
Point 4: Mention the limitations and future works of the developed system elaborately.
Response 4:
Thank you very much for your comments firstly.
At present, our research still stays at the optimization of the algorithm level and has not been implemented on the terminal system. As discussed in Chapter 4 and mentioned in Chapter 5, the limitation of our method is that, firstly, the joint descriptor proposed by our method is based on the three-dimensional features of the structure and does not contain the color information of the building. Therefore, in the face of buildings with similar structures or the same but different colors, it is easy to have mismatches. In addition, in the face of similar or identical buildings, the combination of target-level matching and scene-level matching is also a method to reduce the probability of false matching.
Point 5: Authors should mention the implementation challenges.
Response 5:
Thank you for your comments firstly.
As mentioned above, at present, our work only stays at the optimization of the algorithm level, and there is still a long way to go from the application to the terminal equipment. Therefore, we have not explored the challenges that may be encountered in the implementation of the research.
Point 6: The paper needs intensive proofreading as it contains many long, inconsistent sentences. Besides, the manuscript contains many grammar errors and typos.
Response 6:
Thank you very much for your comments. We carefully proofread the full text again, modified the grammatical errors and typos, and polished the language. We also modify the table to improve readability and aesthetics. The revised part has been marked in the manuscript. Thank you for your comments.

Reviewer 3 Report
The article was interesting and easy to read. However, the usage of space shall be overlooked, e.g., usually between the word and reference there is a space, see rows 15 and 22. Or in some places tehre are spaces in places where not needed (see rows 140-142). For formulas there have to be a desription of symbols used, and it is given after the formula. If one gives reference to the formula, there shall be as well clearly said it. In this case, the bibliography references are as well given in parenthesis, so in some cases it is hard to read.
If one already onece gives the abbreviation, it would be wise to use the abbreviation instead later on.
Figure 4 is not with good quality, it is hard to distinguish gray on the black surroundings, maybe to change the bacround for the models, so the figure becomes more readable?
Author Response
Thank you very much for your comments firstly. According to your suggestion, we optimized the use of spaces in the article, modified and adjusted the interpretation of the formula, strictly corrected the use of abbreviations in the article, and modified the color matching of Figure 4. Thank you for your comments again.
Round 2
Reviewer 2 Report
We appreciated the authors' efforts in manuscript revision, and they did a good job. The following minor concerns need to be discussed and revised carefully before the paper's acceptance.
1 “Figure 1. The comparison diagram of pixel-level matching and object-level matching” description is too short and it needs to be described in details.
2 Introduction section is still weak. An introduction is an important road map for the rest of the paper that should be consist of an opening hook to catch the researcher's attention, relevant background study, and a concrete statement that presents main argument but your introduction lacks these fundamentals, especially relevant background studies. This related work is just listed out without comparing the relationship between this paper's model and them; only the method flow is introduced at the end; and the principle of the method is not explained. To make soundness of your study must include these latest related works. Authors also need to justify the importance of their article and cite all of them to make a critical discussion that makes a difference from others' work.
I Resolution Enhancement for Large-Scale Real Beam Mapping Based on Adaptive Low-Rank Approximation. IEEE Transactions on Geoscience and Remote Sensing, 60, 1-21. doi: 10.1109/TGRS.2022.3202073
II Improved Detection of Buried Elongated Targets by Dual-Polarization GPR. IEEE Geoscience and Remote Sensing Letters, 20. doi: 10.1109/LGRS.2023.3243908
III Dual-Graph Attention Convolution Network for 3-D Point Cloud Classification. IEEE Transactions on Neural Networks and Learning Systems, 1-13. doi: 10.1109/TNNLS.2022.3162301
IV Voids Filling of DEM with Multiattention Generative Adversarial Network Model. Remote sensing (Basel, Switzerland), 14(5), 1206. doi: 10.3390/rs14051206
3 “Figure 5. The structure diagram of different JDEM” should be in “Materials and Methods” section.
Moderate editing of English language required
Author Response
Point 1: “Figure 1. The comparison diagram of pixel-level matching and object-level matching” description is too short and it needs to be described in details.
Response 1:
Thank you very much for your comments firstly. We have made changes according to your suggestion.
Point 2: Introduction section is still weak. An introduction is an important road map for the rest of the paper that should be consist of an opening hook to catch the researcher's attention, relevant background study, and a concrete statement that presents main argument but your introduction lacks these fundamentals, especially relevant background studies. This related work is just listed out without comparing the relationship between this paper's model and them; only the method flow is introduced at the end; and the principle of the method is not explained. To make soundness of your study must include these latest related works. Authors also need to justify the importance of their article and cite all of them to make a critical discussion that makes a difference from others' work.
Response 2:
Thank you very much for your comments.
According to your suggestion, we have rewritten the Introduction of the paper. The updated Introduction is divided into five paragraphs, with the core ideas of each paragraph as follows: Introduce the traditional image-based retrieval BM methods (paragraph 1). Discuss the problems with the application of the above methods and propose the pixel-level cross-dimensional data matching method (paragraph 2). Explain the principle of the pixel-level cross-dimensional data matching method, highlight the issues with this method, and introduce the object-level cross-dimensional data matching method (paragraph 3). Explain the principle and current status of the object-level cross-dimensional data matching method (paragraph 4). Introduce our work, explain the principle and advantages of our proposed method (paragraph 5).
Specifically, the first paragraph focuses on building matching and introduces traditional building matching methods. The second paragraph discusses the problems associated with traditional building matching methods in current applications, such as uneven data distribution and the reliance on large image databases. To address these issues, the pixel-level cross-dimensional data matching method is proposed. The third paragraph explains the principles behind the pixel-level cross-dimensional matching method and describes some of the implementation techniques used. Additionally, some problems associated with this method are highlighted, such as the fuzzy matching caused by sparse point clouds and the inability of loss functions to converge accurately due to differences in data attributes. Therefore, a object-level cross-dimensional data matching method is proposed as a solution. The fourth paragraph provides a detailed explanation of the meaning, principles, and advantages of the proposed object-level cross-dimensional data matching method. The existing object-level matching methods are also discussed, and the differences between their methods and ours are highlighted. Finally, the fifth paragraph explains the principles and advantages of the proposed method.
From image-to-image building matching methods to pixel-level cross-dimensional data matching methods, and then to object-level cross-dimensional data matching methods, there is a progressive relationship between them. The latter method was proposed to address the issues present in the former method. Therefore, related work based on these methods is compared to our proposed method, highlighting clear differences between them.
Based on your comments, we have strengthened the description of the principles behind the different methods. Additionally, we have included references to several recent papers in our introduction, and we appreciate your article recommendations. However, as the articles you recommended were not strongly related to our work, we have not cited them. Nevertheless, we still appreciate your suggestions.
Point 3: “Figure 5. The structure diagram of different JDEM” should be in “Materials and Methods” section.
Response 3:
Thank you very much for your comments firstly. We have made changes according to your suggestion.